# Randomized Spatial PCA (RASP): A computationally efficient method for dimensionality reduction of high-resolution spatial transcriptomics data

Ian K. Gingerich[1,2]*, Brittany A. Goods[2], Hildreth Robert Frost[1]*

**1** Department of Biomedical Data Science, Geisel School of Medicine, Dartmouth College, Hanover, New Hampshire, United States of America, **2** Thayer School of Engineering, Dartmouth College, Hanover, New Hampshire, United States of America

* ian.gingerich.gr@dartmouth.edu (IKG); hildreth.r.frost@dartmouth.edu; rob.frost@dartmouth.edu (HRF)

## Abstract

Spatial transcriptomics (ST) provides critical insights into the spatial organization of gene expression, enabling researchers to unravel the intricate relationship between cellular environments and biological function. Identifying spatial domains within tissues is key to understanding tissue architecture and mechanisms underlying development and disease progression. Here, we present Randomized Spatial PCA (RASP), a novel spatially-aware dimensionality reduction method for ST data. RASP is designed to be orders-of-magnitude faster than existing techniques, scale to datasets with 100,000+ locations, support flexible integration of non-transcriptomic covariates, and reconstruct de-noised, spatially-smoothed gene expression values. RASP itself is not a clustering or domain detection method; cell types and spatial regions are obtained by clustering the RASP PCs, and the effective cluster resolution depends on the K-nearest-neighbor (kNN) graph and a smoothing parameter $\beta$. It employs a randomized two-stage PCA framework and configurable spatial smoothing. RASP was compared to BASS, GraphST, SEDR, SpatialPCA, STAGATE, and CellCharter using diverse ST datasets (10x Visium, Stereo-Seq, MERFISH, 10x Xenium) on human and mouse tissues. In these benchmarks, RASP delivers comparable or superior accuracy in tissue-domain detection while achieving substantial improvements in computational speed. Its efficiency not only reduces runtime and resource requirements but also makes it practical to explore a broad range of spatial-smoothing parameters in a high-throughput fashion. By enabling rapid re-analysis under different parameter settings, RASP empowers users to fine-tune the balance between resolution and noise suppression on large, high-resolution subcellular datasets—a critical capability when investigating complex tissue architecture.

**Data availability statement:** Mouse ovary: Please see the original publication for data download (https://journals.plos.org/plosbiology/article?id=10.1371/journal.pbio.3003193#sec016) DLPFC: We downloaded the raw DLPFC data (slide #151673) from the spatialLIBD website (http://research.libd.org/spatialLIBD/index.html). Mouse Olfactory bulb: The dataset was downloaded from the SEDR publication GitHub repository [69] (https://github.com/JinmiaoChenLab/SEDR_analyses/tree/master/data). Breast Cancer: We downloaded the Breast cancer Xenium dataset from the SubcellularSpatialData R package via ExperimentHub, dataset # EH8567, sample ID 'IDC' (https://www.bioconductor.org/packages/release/data/experiment/html/SubcellularSpatialData.html). Sagittal mouse brain: We downloaded the mouse brain dataset from the Allen Brain Atlas via their abc atlas access Python package. Specifically we followed the MERFISH whole mouse brain spatial transcriptomics (Xiaowei Zhuang) tutorial https://alleninstitute.github.io/abc_atlas_access/notebooks/zhuang_merfish_tutorial.html to access section 3.010 from the 'Zhuang-ABCA-3' dataset. Coronal mouse brain: We downloaded the coronal mouse brain dataset from the publicly available datasets on the 10x website https://www.10xgenomics.com/datasets/visium-hd-cytassist-gene-expression-libraries-of-mouse-brain-he-v4. Simulated datasets and RASP implementation: Please visit our GitHub to access the simulated datasets and their associated models (https://github.com/gingerii/RASP_manuscript). Please visit our GitHub for details on RASP implementation: https://github.com/Goods-Lab/RASP.

**Funding:** This study received support from NIH grants R35GM146586 (IKG, HRF), P20GM130454 (IKG, BAG, HRF) and P30CA023108 (HRF). The funders had no role in study design, data collection and analysis, decision to publish, or preparation of the manuscript.

**Competing interests:** The authors have declared that no competing interests exist.

## Author summary

Spatial transcriptomics (ST) technologies enable unprecedented insights into the spatial organization of gene expression within tissues, yet analysis of these increasingly large and complex datasets remains computationally challenging. We present Randomized Spatial PCA (RASP), a novel, scalable, and computationally efficient dimensionality reduction method tailored for spatial transcriptomics data. Unlike existing methods, RASP can rapidly process datasets with hundreds of thousands of spatial locations and integrates non-transcriptomic covariates to improve biological signal recovery. By combining randomized linear algebra with spatial smoothing, RASP produces spatially informed principal components that support downstream clustering and spatial domain identification across diverse ST platforms, including high-throughput sequencing and in situ imaging technologies. Benchmarking on multiple real and simulated datasets demonstrates that RASP achieves comparable or superior accuracy to state-of-the-art methods while drastically reducing computational time and resource requirements. This efficiency empowers researchers to explore biological questions at multiple spatial resolutions and scales, facilitating robust, high-throughput spatial analysis critical for advancing our understanding of complex tissue architectures.

## 1 Introduction

Spatial transcriptomics (ST) assays measure gene expression at thousands to hundreds-of-thousands of spatial locations within tissue sections, enabling researchers to characterize transcript abundance *in vivo* at resolutions ranging from small cell groups to subcellular structures [1,2]. ST technology facilitates exploration of gene expression's spatial organization, providing insights into cellular microenvironments and functional states of individual cells [3]. Two primary types of ST technologies exist: high-throughput sequencing technologies (sST) and single-molecule fluorescent in situ hybridization or imaging-based technologies (iST) [4,5]. High-throughput sequencing technologies, such as 10x Genomics' Visium, capture mRNA transcripts at multiple sites or spots on a slide. The original 10x Visium platform offers a resolution of 55 μm, with each spot containing between 1–10 cells [6]. Emerging technologies, such as Visium HD, promise higher resolution of 2 μm, approaching single-cell levels, while alternatives like Slide-seq V2 capture transcripts with a resolution of 10 μm, also nearing single-cell resolution [7,8]. iST technologies, which reach single cell resolution, include platforms such as MERFISH/MERSCOPE (Vizgen), seqFISH, osmFISH, COsMx, and 10x Xenium [9–13]. A notable advancement in recent years has been the introduction of iST platforms that support the use of formalin-fixed, paraffin-embedded (FFPE) tissue samples, allowing for use of archival tissue banks, increasing the use of iST technologies [9,12,13]. Additionally, probe sets used in iST technologies have expanded, with systems like 10x Xenium developing probe sets with up to 5,000 individual probes [14], helping to bridge the trade-off between sequencing depth and resolution, allowing for more comprehensive

profiling of gene expression. As costs decrease and technology advances, ST studies will transition from analyzing small sample sizes to extensive investigations [2,13,15–19], generating multi-slide, multi-timepoint ST datasets covering millions of cells across tissues [2]. Larger datasets necessitate development of novel computational tools that leverage both spatial and transcriptomic data efficiently.

ST's rich data allows tissue investigation at multiple biological scales, and computational approaches must adapt. On a cellular level, cell type assignment facilitates spatial distribution visualization, enhancing understanding of cellular structure and function in native tissue [20,21]. At larger scales, identifying tissue domains aids comprehension of complex tissue microenvironments. Identifying cell types and spatial domains is achievable with recently developed computational tools. Popular single-cell RNA sequencing (scRNA-seq) tools like Seurat and Scanpy perform principal component analysis (PCA) and clustering on gene expression data while disregarding spatial context [22,23]. However, methods like BASS and FICT now incorporate spatial context [24,25]. For spatial domain detection, methodologies like BASS, SEDR, GraphST, STAGATE, SpatialPCA, CellCharter, Banksy, NichePCA, and MENDER exist [24,26–33], utilizing different model architectures to embed spatial and gene expression data into low-dimensionality spaces for clustering and domain identification. ST data is often sparse, so de-noising and imputing poorly captured transcripts is a desirable feature, and many spatially-aware methods provide this functionality [34–37].

Yet, current methods have notable limitations. While effective on a single ST slice with few profiled locations, they are computationally intensive, exhibiting slow processing locally, particularly on iST datasets exceeding 100,000 cells. With iST dataset generation increasing, efficient, spatially-aware dimensionality reduction algorithms are necessary. Many existing methods are also inflexible and complex, hindering customization of spatial smoothing or integration of non-transcriptomic covariates, such as those derived from hematoxylin and eosin (H&E) images or protein abundance metrics. Addressing these limitations is crucial for deriving meaningful insights from large-scale ST datasets.

In this paper, we present Randomized Spatial PCA (RASP), a computational method for fast, scalable, and interpretable spatially aware dimensionality reduction. RASP produces spatially smoothed principal components (PCs) and optionally reconstructs de-noised, spatially smoothed gene expression values. Importantly, RASP is not a clustering or domain detection method; cell types and spatial domains are obtained by clustering the RASP PCs with standard algorithms (e.g., Leiden/Louvain or Gaussian mixtures). The effective resolution of the resulting clusters depends on the K-nearest-neighbor (kNN) graph used to define spatial neighborhoods and a smoothing parameter $\beta$ that controls the strength of spatial regularization. RASP employs randomized linear algebra for efficient dimensionality reduction of large datasets [38], supports integration of covariates such as cellular density or sequencing depth, and exposes the kNN and $\beta$ parameters to enable user-controlled smoothing. Tested on six real-world and two simulated datasets (including 10x Genomics Visium/Visium HD and subcellular imaging technologies like Stereo-Seq, MERFISH, and 10x Xenium), pipelines that cluster RASP PCs exhibit tissue-domain concordance comparable to or better than existing methods, with competitive runtimes. This efficiency facilitates systematic exploration of kNN and $\beta$ to balance resolution and noise suppression in large, high-resolution datasets.

## 2 Results

### 2.1 RASP overview

RASP performs dimensionality reduction and spatial smoothing on normalized ST data, with the option to incorporate location-specific covariates. The randomized two-stage PCA optimizes computational efficiency while enabling the flexible and efficient analysis of high-resolution ST datasets. RASP utilizes a sparse matrix representation of the data, conducts dimensionality reduction via randomized PCA, and generates spatially smoothed principal components (PCs) using an inverse distance matrix sparsified by a kNN threshold (Fig 1A). Covariates integration into the latent variables is achieved through an optional second-stage randomized PCA on a matrix with covariates appended to first-stage smoothed PCs. RASP's spatially smoothed PCs are useful for a variety of downstream analyses, including clustering and reduced rank

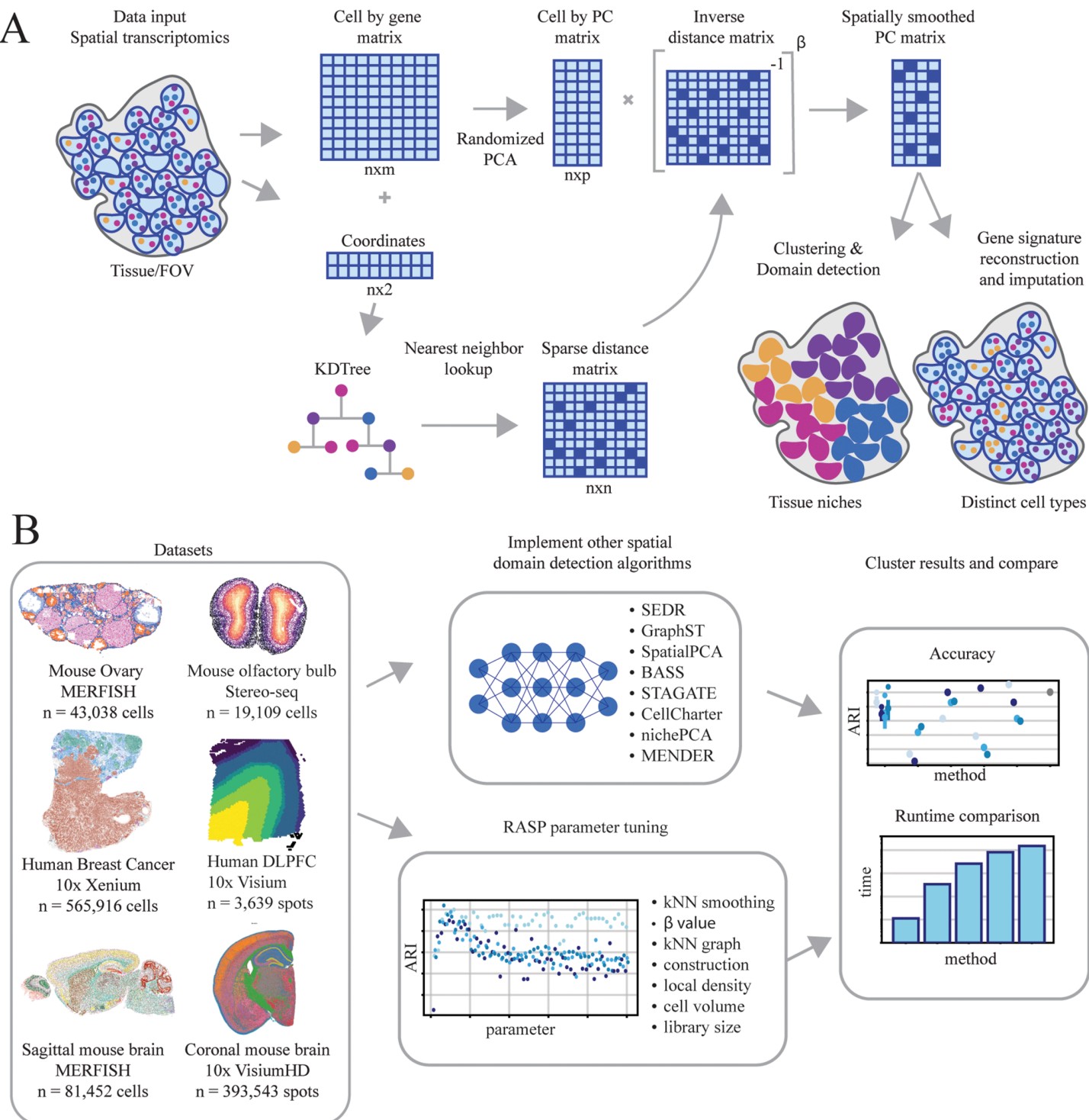

**Fig 1**. **RASP overview. A: Method workflow.** RASP takes as input a cell-by-gene expression matrix and spatial coordinates from a ST experiment. The expression matrix is reduced via randomized PCA and spatially smoothed by a sparse inverse distance matrix. Non-transcriptomic covariates can be added at this stage (not pictured). The output of RASP can be used for downsteam analyses including cell type annotation and spatial domain identification. **B: Evaluation pipeline.** RASP was tested on four ST datasets each generated using a different platform (MERFISH, Stereo-seq, Xenium, Visium) and two synthetic datasets (100 replicates each). RASP results were compared against standard PCA and alternative spatially aware dimensionality reduction models (SEDR, GraphST, SpatialPCA, BASS, STAGATE, CellCharter, nichePCA, and MENDER) for both accuracy and speed.

reconstruction of gene-level expression. See Algorithm 1 and Sect 4.1 for details. Note that by default, RASP utilizes 20 PCs, although this is a user tunable parameter. See Figs O and P in S1 Text for effects of variable PC number on RASP performance.

## 2.2 RASP evaluation on real data

We applied RASP to six publicly available ST datasets generated using diverse techniques and resolutions on human and mouse tissues including the mouse ovary (Vizgen MERFISH technology), mouse olfactory bulb (STOmics Stereo-seq technology), human breast cancer tumor (10x Genomics, Xenium technology), human DLPFC (10x Genomics, Visium technology), and Sagittal mouse brain (Allen institute, Vizgen MERFISH technology), and Coronal mouse brain (10x Genomics, Visium HD technology) (Fig 1B) and Table A in S1 Text. For details regarding field standard preprocessing steps used, such as filtering criteria and normalization, please refer to Sects 4.4.1–4.4.3, and Sect 1.2 in S1 Text, Sect 1.3 in S1 Text.

We evaluated RASP by clustering locations using the spatially smoothed principal components (PCs) output by the algorithm. Cluster quality was assessed through spatial continuity and compactness via CHAOS score (Sect 1.3 in S1 Text), spatial autocorrelation by Moran's I (Sect 1.4 in S1 Text) and the Adjusted Rand Index (ARI) (Sect 1.2 in S1 Text) compared to ground truth annotations. We conducted a sensitivity analysis on key RASP parameters, focusing on the kNN distance threshold for sparse matrix construction and the $\beta$ power parameter (see Fig 1A). This analysis used a one-parameter-at-a-time strategy to investigate parameter impacts on clustering outcomes.

Performance comparisons were made between RASP and eight other spatially informed dimensionality reduction or spatial domain detection methods: GraphST [26], SEDR [27], SpatialPCA [28], STAGATE [29], CellCharter [30], nichePCA [32],MENDER [33], and the joint cell type and domain inference tool BASS [24], as well as performing randomized PCA. For each comparison, we adhered to the respective GitHub tutorials to run these methods for spatial domain and cell type detection. Clustering of latent dimensions found by each method was performed using Mclust [39], Louvain [40], Leiden [41], and Walktrap [42] algorithms, where appropriate. For datasets with ground truth annotations, the ARI values presented are based on a default range of parameters outlined in Table 1.

## 2.3 RASP identifies biologically relevant structures in heterogeneous mouse ovary MERFISH data

We selected this MERFISH dataset for its complex tissue structure and cellular resolution, featuring spatially distinct and heterogeneous cell types. Obtained from Huang et al., [15] it includes ground truth cell type annotations based on histology and differential gene expression (Fig 2A). Annotations include luteal, luteinizing mural, granulosa, theca, and stromal cells, with clear spatial organization, alongside endothelial, epithelial, and immune cells, which are heterogeneously distributed across the tissue [43], making automatic annotation challenging.

For this dataset, RASP was set at the recommended 'cell type' default parameters (Table 1), using Louvain clustering achieved the highest ARI of 0.69, followed by standard randomized PCA (ARI of 0.58) and SEDR (ARI of 0.57) (see Fig 2A and 2C). We employed three different clustering methods to assign labels to cells in the dataset. The Louvain and Leiden algorithms performed best, while Mclust the worst. Note that BASS utilizes Mclust as a prior in its Bayesian framework but does not support a user-defined clustering algorithm.

**Table 1**. **RASP parameter suggestions.**

| Technology | Label type | $\beta$ | kNN Default | kNN Range | Moran's I Value | CHAOS Value |
|---|---|---|---|---|---|---|
| Visium | small/cell type | 2 | 1 | 1-2 | smaller is better | larger is better |
| Visium | spatial domain | 0 | 5 | 3-10 | larger is better | smaller is better |
| All others | small/cell type | 2 | 10 | 2-20 | smaller is better | larger is better |
| All others | spatial domain | 0 | 50 | 30-100 | larger is better | smaller is better |

PLOS Computational Biology

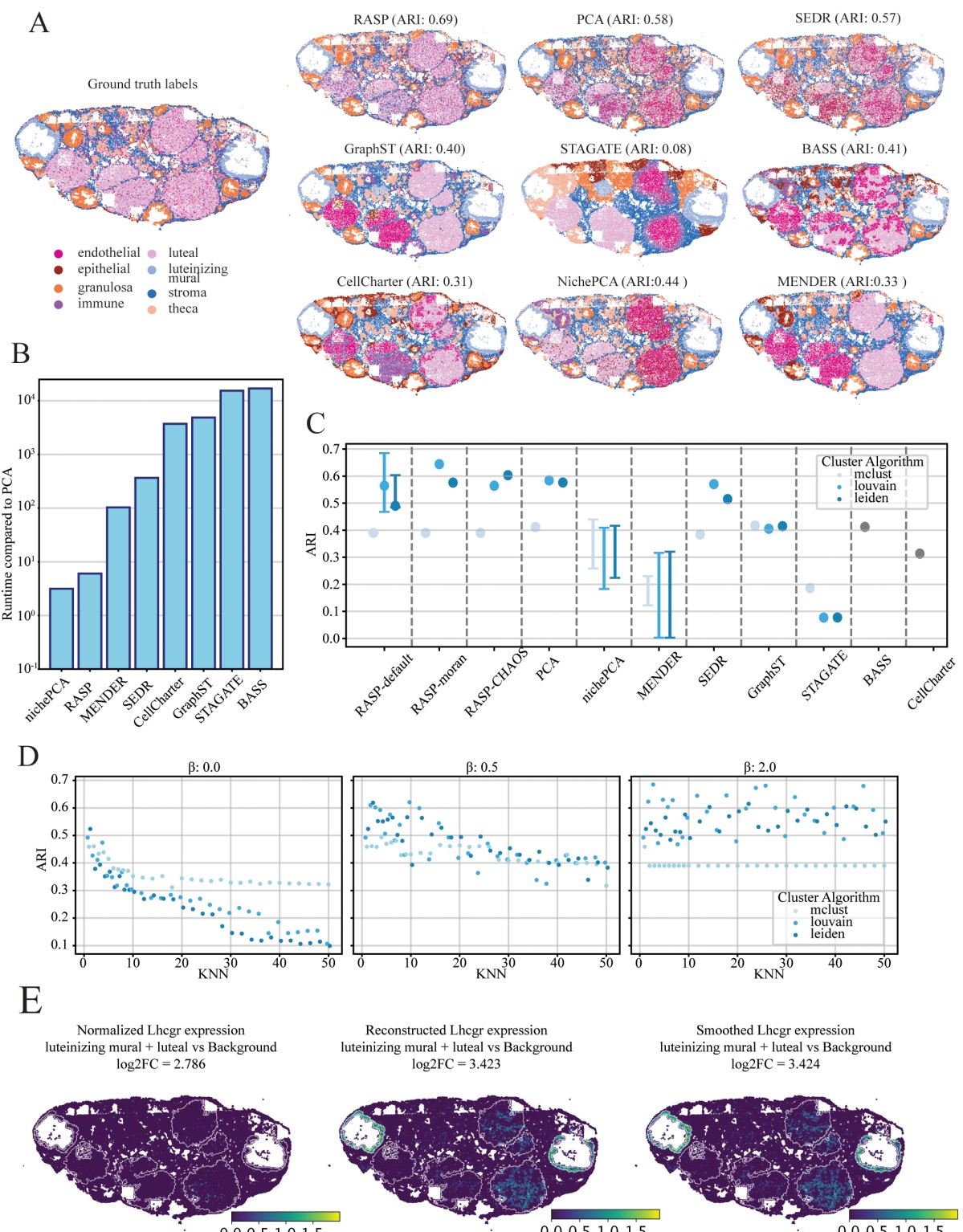

**Fig 2. Overview of mouse ovary analysis using Vizgen MERFISH data. A: Cell Type Annotation Comparison** — Ground truth cell type labels (top left) alongside predictions from RASP, normal PCA, and other methods, illustrating accuracy of spatial mapping. **B: Runtime Performance** — Comparison of computational runtime across methods, highlighting RASP efficiency relative to standard randomized PCA. **C: Clustering Accuracy (ARI) Across Methods** — Adjusted Rand Index values for all methods, color-coded by clustering algorithm; shows full ARI range at default RASP parameters

(kNN = 2–20, $\beta = 2$) and the default parameter single-point performance (kNN=10, $\beta = 2$). RASP-moran and RASP-CHAOS indicate ARI when using label-agnostic metrics for parameter selection. **D: Effect of $\beta$ on ARI** — ARI values for RASP with inverse distance weighting raised to $\beta = 0$ (left), 0.5 (middle), and 2 (right), plotted against kNN values; colors indicate clustering algorithm used. **E: Spatial Expression Patterns of Lhcgr** — Normalized expression (left), reduced rank reconstructed expression (center), and spatially smoothed reconstructed expression (right). White lines depict luteinizing mural and luteal cell compartment boundaries.

Relative runtimes show RASP is comparable to nichePCA and one to three orders-of-magnitude faster than other methods on this dataset (Fig 2B). RASP performance diminishes with increasing kNN thresholds for PC smoothing; high-lighted by decreasing ARI scores at larger kNN values (Fig 2D). This result is not unexpected for cell type annotation, but it is noteworthy that the local smoothing employed by RASP in this task, essentially borrowing information from just a few neighboring cells, has a marked increase in performance over normal randomized PCA for clustering-based label assignment.

Using RASP-based reduced rank reconstruction of gene expression profiles, we analyzed the spatial signature of the luteinizing hormone/chorionic gonadotropin receptor (Lhcgr) given its essential role in ovarian biology and ovulation (Fig 2E). Lhcgr is a G-protein coupled receptor that binds luteinizing hormone and chorionic gonadotropins, is expressed on luteal and luteinizing mural cells [44–46]. Originally, gene expression showed minimal spatial patterning (Fig 2E, left plot). After PCA-based reduced rank and RASP-based reconstruction, expression is localized to luteinizing mural and luteal cells (Fig 2E, middle and right plots). When comparing normalized expression log-2 fold-change (log2FC) between luteinizing mural and luteal cells vs all other cells, both PCA-based and RASP-based reduced rank reconstruction results in higher mean expression (localization) to these regions. For this example, we utilized the distances to only 2 kNN, which results in very similar log2FC values between normal PCA-based and RASP-based reconstruction.

## 2.4 Incorporation of covariates improves RASP annotation of mouse ovary ST data

A key feature of RASP is support for integrating non-transcriptomic covariates into reduced dimensions. The two-stage PCA approach used by RASP offers benefits relative to a single PCA on merged transcriptomic and non-transcriptomic variables: It prevents covariate signal dilution within the high-dimensional transcriptomic data, and decouples spatial smoothing of expression data from covariates. To RASP covariate integration, we explored three covariates on analysis of the mouse ovary data: local cell density, transcriptomic library size, and cell volume. For each covariate, we performed parameter sweeps over the kNN parameter using three RASP configurations: 1) standard RASP without covariates (i.e., the configuration used to generate the results in Fig 2) 2), single-stage RASP executed on the merge of ST expression data and covariates, and 3) two-stage RASP with the covariates added to the output from the first stage (see Fig 3A (bottom)). For these experiments, the $\beta$ parameter was held constant at $\beta = 2$, the optimal value for this dataset (see Figs 2D, and A in S1 Text). For local density, spatial smoothing was applied to the PC matrix, but not the local density variable, as local density inherently contains spatial information. Library size and cell volume covariates where smoothed by the same inverse distance weighting as the PCs matrix, prior the second round of dimensionality reduction. In addition, we explored the addition of local cell density for the breast cancer dataset, using the same experimental design described above, with the RASP kNN held constant at 60. The results were less striking than the ovary dataset, with both the one and two stage versions improving the clustering across some of the density values for Louvain clustering, and generally reducing performance when using Leiden or mclust algorithms (Fig F in S1 Text).

All three covariates impacted ARI values, with clustering algorithm determining the magnitude and direction of the ARI change. The ARI using Mclust improved regardless of covariate added, with single-stage RASP equal to or outperforming the two-stage version in most cases (see Fig 3B (left column)). Mclust ARI without covariates was 0.39 improving to 0.45 and 0.5 with covariates. ARI with Leiden was highest for two-stage RASP in most cases, while the single-stage version

PLOS Computational Biology

# A

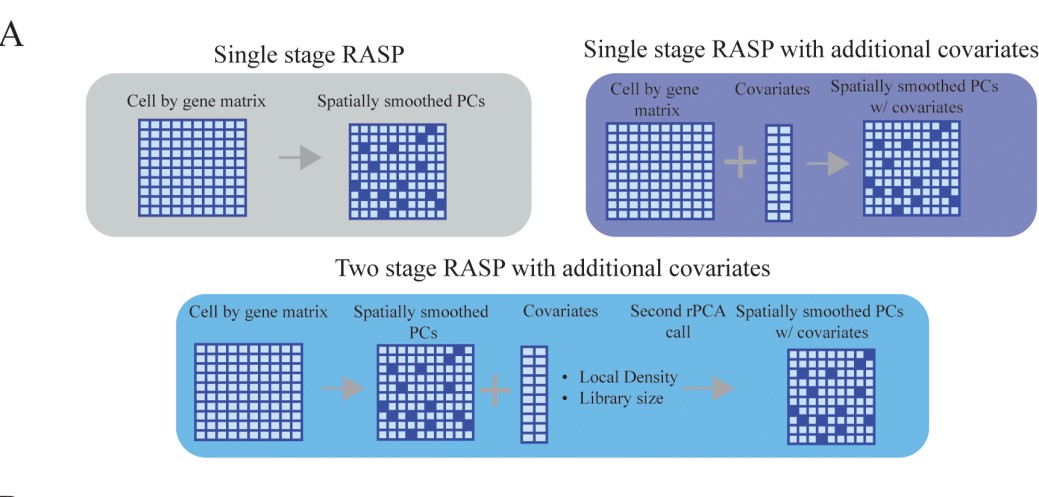

# B

Fig 3. **Covariate analysis (Mouse ovary dataset) for the cell type clustering case. A: RASP Model Architectures** — Illustrations of the base-line RASP model without covariates (top left), the single-stage RASP model incorporating one covariate (top right), and the two-stage RASP model architecture that integrates an additional covariate for improved clustering. **B: Clustering Performance with Covariate Adjustment** — Adjusted Rand

Index (ARI) comparisons across models without covariates, single-stage RASP, and two-stage RASP, plotted against varying kNN values. Columns represent different clustering algorithms, while rows show the impact of different covariates: local cell density (top), cell library size (middle), and cell volume (bottom).

was covariate dependent, with the addition of local density decreasing ARI values, and addition of library size and cell volume having mixed effects (see Fig 3B (right column)). Notably, Leiden clustering with cell volume achieved a top ARI of 0.71, the highest attained for this dataset. Louvain- classification was hindered by local density additions, with ARI values dropping across kNN values, while affects of library size and cell volume fluctuated with kNN value. Overall, these results indicate that integration of covariates can enhance RASP performance, but covariate selection, clustering algorithm, spatial smoothing require careful consideration. Importantly, two-stage RASP avoids covariate dilution offering more control over spatial smoothing.

## 2.5 RASP enables the investigation of tissue domains at multiple scales

To investigate RASP's capacity to resolve tissue structure across a hierarchy of physical scales, we applied it to the Allen Institute's adult mouse brain atlas [47],[1]. This dataset provides comprehensive cell type annotations as well as multiple hierarchical levels of cortical and subcortical brain region annotations, allowing for the assessment of both fine-grained (cell type) and coarse-grained (anatomical region) clustering performance. We compared RASP followed by clustering to the ground truth cell type labels and broad brain region divisions (Fig 4A and 4D). For the cell typing task, RASP achieved an ARI of 0.77, substantially surpassing other state-of-the-art clustering methods, using cell type parameter defaults ($\beta = 2.0$, kNN=2–20; Fig 4B, and 4C). For the higher-level regional annotation task, RASP achieved an ARI of 0.55, outperforming other approaches with parameters designed for large-scale spatial domains ($\beta = 0$, kNN=50–100; Fig 4E and 4H). See Sect 2.11 for further discussion on RASP's multi-scale ability.

## 2.6 RASP shows comparable performance to alternative methods for cortical layer identification using human DLPFC Visium data

The dataset utilized for this analysis was generated using the 10x Visium platform by the LIBD human Dorsolateral Prefrontal Cortex (DLPFC) project [48] and includes manual cortical layer annotations. This dataset serves as a gold standard benchmark for spatial domain detection algorithms due to its clear neuroanatomical boundaries providing reliable ground truth and a small number of profiled tissue locations facilitating evaluation of computationally complex algorithms. For benchmarking, we selected slice #151,673 from the LIBD dataset, which consists of seven spatial domains including six cortical layers and white matter [48].

For this dataset, most of the tested methods achieved competative ARI values between 0.55 and 0.63 (Fig 5A and 5D). Importantly, RASP has comparable computational cost to randomized PCA but with substantially higher ARI. RASP was one to three orders-of-magnitude faster than other methods, highlighting its efficiency and accuracy (Fig 5B). For optimal RASP performance, the default for visium 'large' of kNN 3–10 was most effective; larger kNN thresholds reduced scores, as they smooth across cortical layer boundaries, creating overly large domain identification (Fig 5D). Interestingly,all $\beta$ values generated similar ARI vs. kNN threshold trends. The Mclust algorithm was relatively invariant to both the $\beta$ and kNN threshold parameters, showing consistent performance across parameter space. Louvain and Leiden algorithms dropped in performance as the kNN threshold increased, regardless of the $\beta$ value, with $\beta = 2$ performing marginally better (Figs 5, and C in S1 Text).

To demonstrate the use of RASP-based reduced rank reconstruction, the TMSB10 (thymosin beta 10) was reconstructed [44] (Fig 5E). The RASP-based reconstruction of TMSB10 is better localized to layer 5 than both normalized

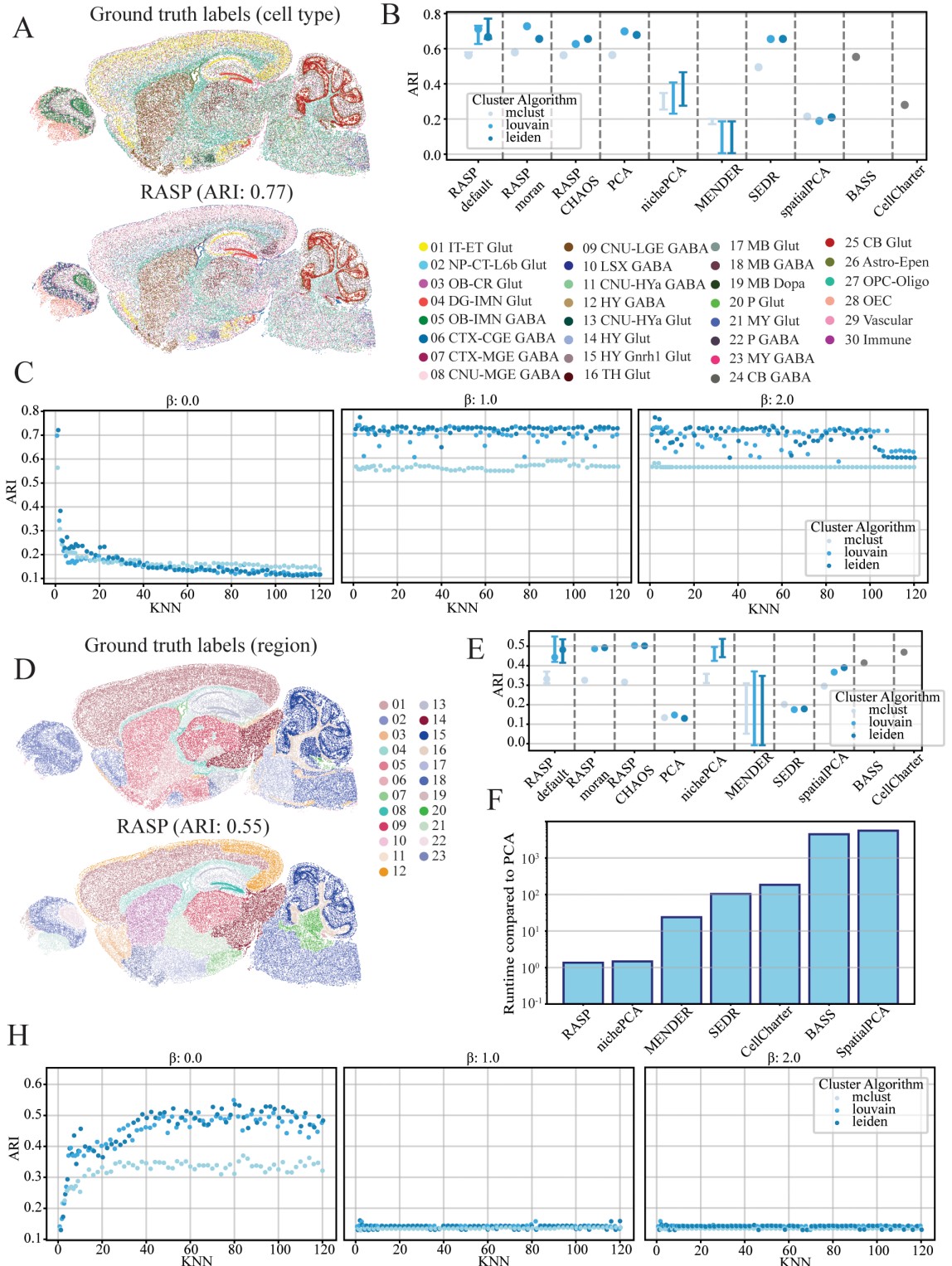

**Fig 4**. **RASP performance across different scales in the Mouse Sagittal brain dataset. A: Cell Type Annotations** — Ground truth cell type labels (top) alongside predicted labels identified by RASP (bottom), demonstrating cell-level classification accuracy. **B: Cell Type Clustering Accuracy** — Adjusted Rand Index (ARI) values for all methods based on cell type annotations in **A**, with colors indicating clustering algorithms. Displays full ARI range at default RASP parameters (kNN = 2–20, $\beta$ = 2) and single-point ARI for default parameters (kNN = 10, $\beta$ = 2). RASP-moran and RASP-CHAOS

indicate ARI when using label-agnostic metrics for parameter selection. **C: Effect of** $\beta$ **on Cell Type Clustering** — ARI values for RASP with inverse distance weighting raised to $\beta = 0$ (left), 1 (middle), and 2 (right), plotted against kNN values; colors denote clustering algorithm. **D: Region Annotations** — Ground truth spatial brain region labels and corresponding regions identified by RASP. **E: Region-Level Clustering Accuracy** — ARI quantification similar to **B**, but computed using region annotations from **D**. Shows full ARI range at default RASP parameters (kNN = 50–100, $\beta = 0$) and single-point ARI performance (kNN = 50, $\beta = 0$). RASP-moran and RASP-CHAOS indicate ARI when using label-agnostic metrics for parameter selection. **F: Runtime Comparison** — Computational runtime of all methods relative to randomized PCA, highlighting efficiency. **H: Effect of** $\beta$ **on Region-Level Clustering** — ARI values computed on region annotations (**D**) with inverse distance weighting raised to $\beta = 0$ (left), 1 (middle), and 2 (right), plotted against kNN values; colors indicate clustering algorithms.

expression and PCA-based reduced rank reconstructed expression, as indicated by a higher log2FC in the region compared to all other spots, highlighting RASP's utility for spatially informed gene-level analyses.

## 2.7 RASP effectively characterizes tissue structures in mouse olfactory bulb Stereo-seq data

The mouse olfactory bulb analysis used a dataset generated with Stereo-seq technology, capturing subcellular tissue resolution, providing a valuable benchmark. The olfactory bulb consists of distinct cortical layers: meninges, olfactory nerve layer (ONL), glomerular layer (GL), external plexifrom layer (EPL), mitral cell layer (MCL), internal plexiform layer (IPL), granule cell layer (GCL), and the rostral migratory stream (RMS) [49,50]. RASP, SEDR, GraphST, CellCharter, MENDER, and nichePCA produced clusters that effectively align with olfactory bulb layers (Fig 2A). In contrast, STAGATE, BASS, randomized PCA, and SpatialPCA were unable to characterize tissue structures in the core of the olfactory bulb.

RASP demonstrated computational costs comparable to randomized PCA and two to three orders-of-magnitude faster than the other methods (Fig 6B). Due to lack of ground truth annotations cluster quality was assessed using Moran's I and CHAOS score (Fig 6C and 6D and Algorithms G and H in S1 Text). RASP exhibited the highest Moran's I and lowest CHAOS scores among evaluated methods, with Louvain and Leiden clustering producing favorable outcomes. Analysis of different kNN thresholds revealed increased kNN values resulted in higher Moran's I values and lower CHAOS scores, though beyond a kNN of 50, clusters deviated from anatomical layers, producing large homogeneous structures that, while rewarded by the CHAOS score, lack biological plausibility (Fig 6D). RASP achieved the highest Moran's I value and lowest CHAOS score with low $\beta$ values ranging from 0–0.5. As the $\beta$ parameter increased past this point, Moran's I decreased and the CHAOS score increased, indicating less optimal clustering solutions (Fig 6 and E in S1 Text).

RASP-based reduced rank reconstruction performance was explored using the Doc2g gene, which acts upstream of neuro- transmitter secretion and locates in membrane and presynapses of neurons [51,52]. Within the olfactory bulb, Doc2g is localized to the EPL and IPL. While normalized expression of Doc2g in the original dataset appears sparse and noisy, RASP-based reconstruction is clearly localized to the EPL and IPL, showing greater spatial organization than standard PCA-based reconstruction Fig 6E).

## 2.8 RASP efficiently scales to large human breast cancer Xenium data while maintaining accurate tissue domain detection

To assess RASP on large, high-resolution data, we downloaded the 10x Xenium human breast cancer dataset and annotations from the R package SubcellularSpatialData [53]. This dataset originates from a publicly available 10x Genomics dataset of re-sectioned FFPE human breast tissue infiltrating ductal carcinoma *in situ*. Annotations are based on H&E stained tissue accompanying the Xenium data. Due to it's size, over half a million cells, competing domain detection methods encounter out-of-memory errors, requiring over 1 TB of RAM- unrealistic for many researchers, even with institutional high-performance computing environments. Given these limitations, we compared RASP against randomized PCA. RASP substantially outscored PCA (ARI 0.63 vs ARI 0.25 ) (Fig 7A and 7B).

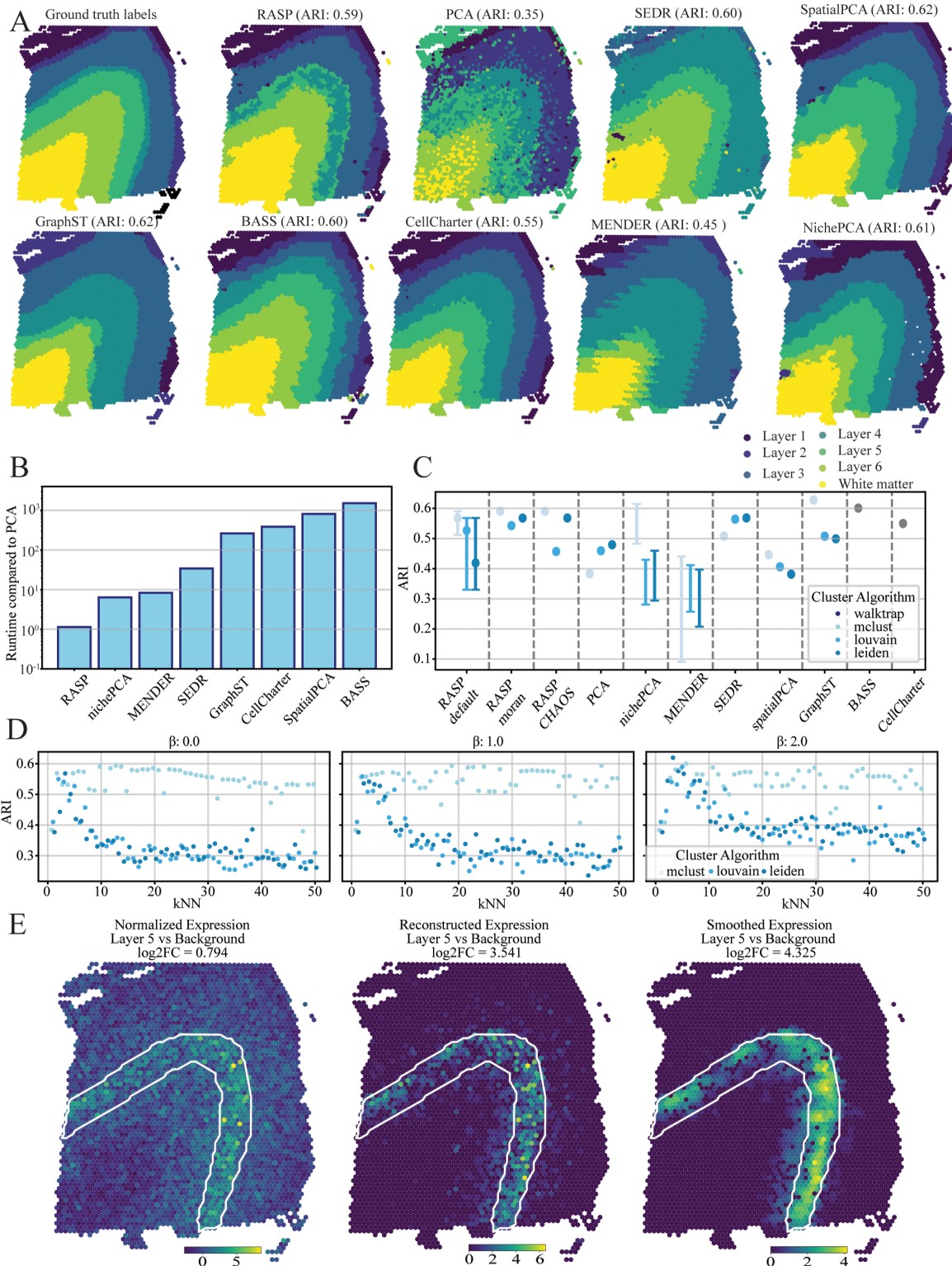

**Fig 5. Human dorsolateral prefrontal cortex (DLPFC) spatial transcriptomics analysis, 10X Visium data. A: Cortical Layer Identification** — Ground truth cortical layer annotations (left) compared to spatial domains identified by RASP, PCA, and other competing methods, demonstrating spatial domain delineation accuracy. **B: Runtime Comparison** — Computational runtimes for all methods relative to randomized PCA, highlighting relative efficiency. **C: Clustering Accuracy (ARI) Across Methods** — Adjusted Rand Index (ARI) values for methods with colors indicating clustering

algorithms. Shows the full range of ARI values at default RASP parameters (kNN = 3–10, $\beta = 0$) alongside the single-point ARI performance at the default parameter (kNN = 5, $\beta = 0$). RASP-moran and RASP-CHAOS values reflect ARI from label-agnostic metric-based parameter selection. **D: Impact of $\beta$ on Clustering Performance** — ARI values for RASP with inverse distance weighting raised to $\beta = 0$ (left), 1 (middle), and 2 (right), plotted against varying kNN values; colors denote clustering algorithms. **E: Spatial Expression of TMSB10** — Normalized TMSB10 expression (left), reduced rank reconstructed expression (center), and spatially smoothed reconstructed expression (right), all plotted on tissue sections. White lines indicate the border of cortical layer 5.

The default parameters for large domains (kNN 30-100, $\beta = 0$) resulted in highly variable ARI values (Fig 7B and 7D). For this dataset, using the label agnostic Moran's I and CHOAS score metrics to determine optimal parameters was favorable, with Leiden and Louvain clustering resulting in better outcomes than Mclust. It is important to consider the computational burden of clustering algorithms, especially given the lengthy runtimes associated with large datasets. Mclust can be three to four times as more computationally expensive as Louvain or Leiden, but if testing multiple resolutions to achieve a desired number of clusters, the three algorithms are comparable. Similar to the olfactory bulb dataset, smaller $\beta$ values (0–0.5) produce better results, with ARI values dropping considerably for all clustering algorithms and all kNN thresholds at higher $\beta$ values (Figs 7D and GA in S1 Text).

To compare RASP against other domain detection methods, we down-sampled the breast cancer dataset to 10% of its original size and compared the ARI values against RASP (Fig F in S1 Text). RASP outperformed all other methods, which generated ARI values comparable to normal PCA. We do not provide a runtime quantification against RASP for this analysis as RASP operated on the full dataset while the other methods did not.

To explore RASP-based reduced rank reconstruction, we compared normalized and reconstructed expression signatures for two genes: FAM3B and CD52. FAM3B, a metabolism-regulating signaling molecule, is predicted to enable cytokine activity [54]. The FAM3 family genes play crucial roles in the malignant development of various human cancers and are largely localized to the DCIS-annotated region in this dataset [55,56]. RASP-based reduced rank reconstruction of FAM3B shows increased localization to the DCIS region. However, because this gene exhibits sparse expression throughout invasive tumor regions, the log2 fold change of expression in the DCIS region compared to background is lower in the RASP-based reconstruction than in the PCA-based reconstructed case. This effect arises because the large kNN value used during smoothing borrows signal from many neighboring cells, diluting the localized expression contrast.

CD52 is a cell surface glycoprotein involved in immune cell signaling and regulation [57]. CD52 is predominantly expressed in immune cells and plays a critical role in modulating immune responses [58]. In this dataset, CD52 is enriched in the regions annotated as immune cells. RASP-based reduced rank reconstruction of CD52 shows expression spreading into infiltrating immune cells located within the invasive tumor. Similar to FAM3B, this spreading dilutes the localization signal, resulting in a lower log2FC of CD52 expression in the immune cell region compared to background in the RASP-based reconstruction relative to the normalized expression. However, the visualization also highlights hotspots of expression towards the bottom of the tissue, highlighting potential immune cell compartments that were not properly annotated by the pathologist.

Together, these findings highlight the unique capacity of RASP to deliver accurate spatial domain detection at scale, even for ultra-high-resolution datasets beyond the reach of competing methods. By enabling rapid parameter exploration and robust clustering on the entire Xenium dataset—without the need for sketching or specialized hardware—RASP empowers researchers to retain biological complexity and make high-confidence discoveries in challenging clinical specimens.

### 2.9 RASP can efficiently analyze Visium HD without the need for downsampling

We applied RASP to the publicly available 10x Visium HD mouse coronal brain dataset, which was aggregated to 8 μm spots, encompassing over 393,000 spots (393,543) and profiling 19,059 genes. To evaluate RASP's performance, we

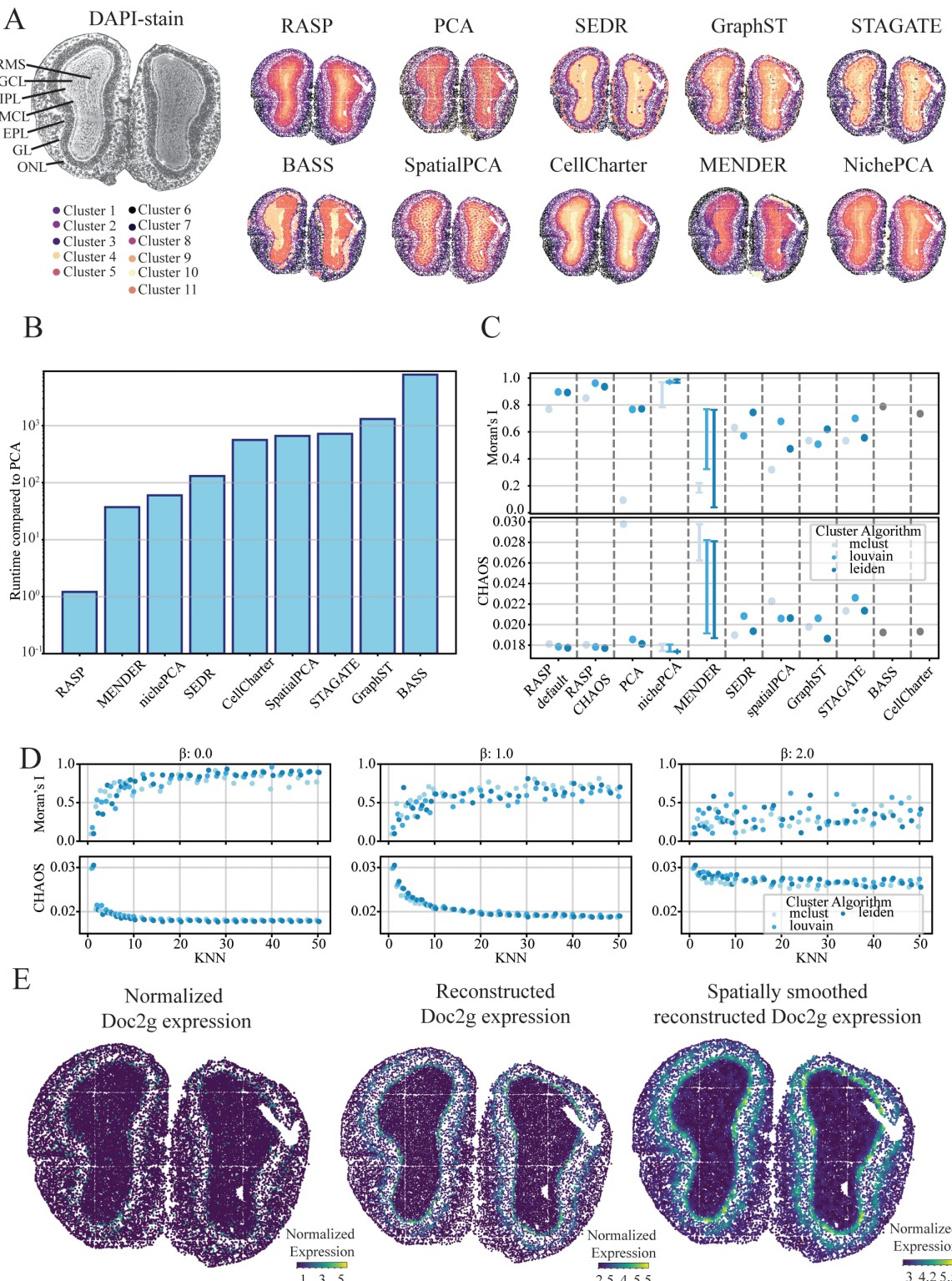

**Fig 6**. **Spatial domain analysis of the mouse olfactory bulb using STOmics Stereo-seq data. A: Laminar Structure Identification** — Ground truth cortical laminar structure (top left) alongside spatial domains identified by RASP, PCA, and other methods, demonstrating domain detection accuracy. **B: Runtime Performance** — Comparison of computational runtimes for all methods relative to normal PCA, highlighting efficiency differences.

**C: Spatial Autocorrelation Metrics** — Moran's I (top) and CHAOS (bottom) scores for all methods, with colors indicating clustering algorithms. Default and best scores are reported for RASP with default parameters (kNN = 30–100, $\beta = 0$). **D: Effect of $\beta$ on Spatial Metrics** — Moran's I (top) and CHAOS (bottom) scores for RASP as inverse distance weighting is raised to $\beta = 0$ (left), 1 (middle), and 2 (right), plotted against varying kNN; colors indicate clustering algorithm. **E: Spatial Expression of Doc2g** — Normalized expression (left), reduced rank reconstructed expression (center), and spatially smoothed reduced rank reconstructed expression (right) of Doc2g plotted on tissue.

compared both "cell type" and "spatial domain" parameterizations on this large-scale dataset against several established methods.

We observed that the performance trends and parameter sensitivities for Visium HD closely mirror those from the mouse olfactory bulb Stereo-seq dataset (see Sect 2.7), indicating consistent behavior of RASP across different spatial transcriptomics platforms and tissue types. Due to the computational burden associated with Visium HD's large dataset, RASP was run on the full dataset without downsampling. In contrast, the other benchmark methods were applied to a downsampled version of the data, reduced to 10% of the original spots, to allow feasible runtimes while maintaining comparative context. These results are detailed in Figs X-Y in S1 Text, showcasing the comparative performance and parameter exploration across all methods.

## 2.10 Simulation analysis

In addition to evaluating RASP on real data, we applied RASP and comparison techniques to simulated ST datasets generated using the **R** package **SRTsim** [59] for two different tissue models with 100 replicates each. The first model, the *Stripes* model, represents a simple tissue with a laminar structure of eight tissue domains (see Fig I in S1 Text). The second model, the *Dots* model, represents a complex tissue with heterogeneous circular regions scattered across the tissue, and a background cell type (Fig L in S1 Text).

RASP outperformed all other methods in computational speed and achieved comparable using default parameters. For the *Stripes* model, the 'spatial domain' parameter settings (kNN 30–100) is needed to correctly identify the laminar tissue-spanning domains (Fig I in S1 Text). For clustering *Stripes* data, Leiden and Louvain were found to be optimal, followed by Walktrap and Mclust. Similar to the olfactory bulb and breast cancer datasets, smaller $\beta$ values (0–0.5) produced better results, with ARI values dropping for all clustering algorithms and kNN thresholds at higher $\beta$ values (Fig I in S1 Text).

For the *Dots* model, the 'cell type' parameter setting (kNN 2–20) is needed to identify the regions in this dataset (Fig L in S1 Text). For clustering *Dots* data, Walktrap was optimal, followed by Mclust, Leiden and Louvain. Similar to the Ovary dataset, larger $\beta$ values between 1.5–2.0 were found to produce better results, with ARI values dropping considerably for all clustering algorithms at lower $\beta$ values (Fig L in S1 Text). Similar to the *Stripes* model, RASP outperformed all other methods in speed and produced accurate label predictions across its default parameters.

## 2.11 RASP facilitates scalable, multi-resolution exploration through computational efficiency

A unique advantage of RASP is its computational efficiency, which enables the biologically meaningful exploration of ST data at multiple scales. Across a diverse collection of ST datasets, spanning multiple tissues and profiling technologies, RASP consistently outperformed or matched other state-of-the-art methods while being orders-of-magnitude faster. In the complex, cellular-resolution MERFISH ovary dataset, RASP was one-to-three orders-of-magnitude faster than competing algorithms (Fig 2B). This speedup persisted even when annotations were spatially heterogeneous and challenging to resolve, such as immune and endothelial populations. SEDR was approximately one order-of-magnitude slower, and BASS nearly four orders-of-magnitude slower, highlighting the impact of algorithmic efficiency when analyzing large, high-resolution ST data. For datasets at anatomical or intermediate scale (e.g, the DLPFC, mouse olfactory bulb, breast cancer, and mouse brain datasets), RASP's efficiency enabled practical multi-scale analysis that would otherwise be prohibitive due to memory and runtime constraints. Notably, in the breast cancer dataset containing over half a million cells,

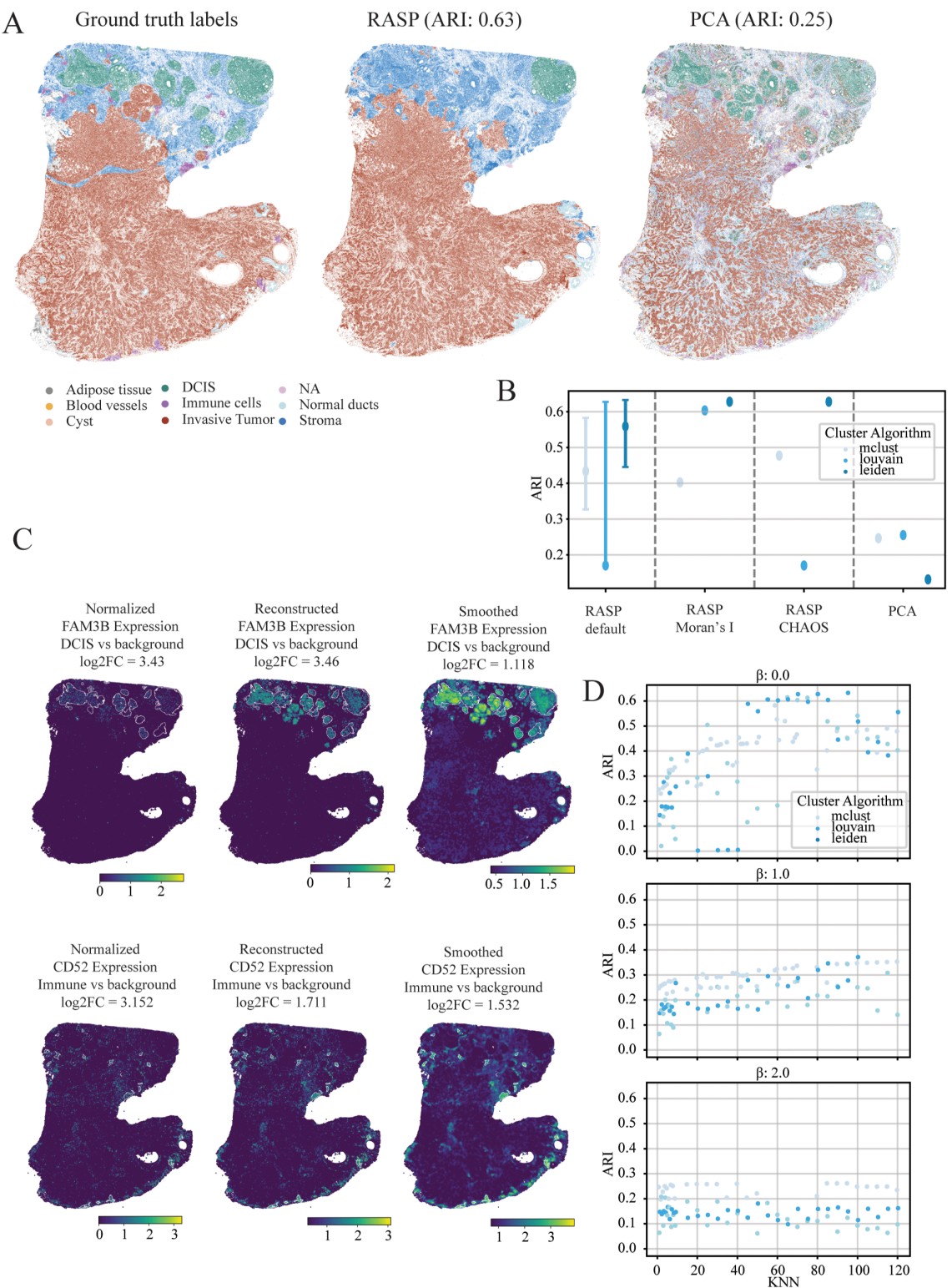

**Fig 7. Spatial domain analysis of human breast cancer tissue using 10x Xenium data. A: Ground Truth and Predicted Spatial Domains** — Ground truth spatial domain annotations (top left) compared to spatial domain predictions from RASP and normal PCA, illustrating method performance in domain delineation. **B: Clustering Accuracy Across Methods** — Adjusted Rand Index (ARI) values for all methods, with colors representing different clustering algorithms. Displays full range of ARI scores at default RASP parameters (kNN = 30–100, $\beta = 0.0$), along with the single-point

performance at default parameter values (kNN = 50, $\beta$ = 0.0). RASP-moran and RASP-CHAOS indicate ARI computed using label-agnostic metrics for parameter selection. **C: Spatial Expression of FAM3B and CD52** — Normalized expression (left column), reduced rank reconstructed expression (center column), and spatially smoothed reduced rank reconstructed expression (right column) of FAM3B and CD52 plotted on tissue sections. White boundaries indicate DCIS (top row) and Immune cell compartments (bottom row). **D: Influence of Smoothing Parameters on Clustering** — ARI values for RASP plotted against smoothing distance (top) and kNN (right), with colors indicating different clustering algorithms.

RASP readily analyzed the full dataset, a task that caused other methods to fail due to memory limits (requiring over 1 TB of RAM to train the model).

The computational speed of RASP allows users to systematically and interactively explore the effects of the kNN threshold and $\beta$ exponent, supporting robust identification of optimal domain solutions tailored to the driving biological questions that motivated the experiment. For instance, in the ovary dataset, local smoothing (small kNN) and larger $\beta$ yielded superior performance for cell type annotation, while higher kNN was necessary to delineate tissue-scale domains in breast cancer. Because RASP's runtime is so low relative to alternative techniques, researchers are not forced to compromise on parameter search or multi-scale analysis, but can instead comprehensively interrogate tissue structure at all relevant spatial scales. Importantly, the rPCA embedding need only be computed once: subsequent parameter sweeps simply rebuild the sparse distance matrix, apply the inverse-distance weighting to the fixed rPCA coordinates, and rerun clustering. To guide choice of optimal settings in a label-free manner, users can compute spatial-autocorrelation statistics (e.g. Moran's I) or boundary-coherence measures like the CHAOS score for each domain solution, and then select the (kNN, $\beta$) combination that maximizes these metrics. This workflow makes it trivial to identify biologically meaningful multi-resolution domains without ever repeating the costly dimension-reduction step. In sum, RASP's computational efficiency not only matches or exceeds state-of-the-art clustering accuracy, but also fundamentally enables the multi-resolution, exploratory analyses needed to reveal the full complexity of spatial tissue architectures.

### 2.12 PC number impacts RASP performance

For the real ST analyses detailed in Sects 2.3–2.8, we computed 20 spatially-smoothed PCs using RASP and performed location-based clustering on the components. Determining the optimal rank for dimensionality reduction is a long-standing problem in statistical analysis and ongoing topic of debate within the bioinformatics community for scRNA-seq and ST data. In practice, analyses typically use a default of between 20 and 50 PCs. Tools like **Scanpy** and **Seurat** default to retaining 30 and 50 PCs respectively, serving as a useful benchmarks.

To evaluate robustness, we ran RASP using the best-performing parameters across the DLPFC, mouse ovary, and breast cancer datasets while retaining 100 PCs. We iteratively clustered each dataset using an increase number of retained PCs, from 5–60, to understand the correlation between PC count and cluster label accuracy (Fig O in S1 Text). For the DLPFC dataset, stable ARI values were observed with increasing PC number for the Louvain, Leiden, and Mclust clustering methods, and an increase for Walktrap. This invariance suggests these clustering models describe data distribution well regardless of dimensions. Conversely, in the mouse ovary dataset, we noted an upward ARI trend with more PCs for the Louvain and Leiden clustering methods. This behavior is consistent with the ground truth annotations of cell types present in this tissue, as higher order PCs are adept at capturing the nuanced details required for accurate cell typing (Fig P in S1 Text). Interestingly, Mclust improved at the lowest PC count, likely because Mclust assumes clusters have consistent size and shape, which isn't the case here. In the breast cancer dataset, Louvain and Leiden algorithms showed increased ARIs up to 30 PCs, and then values decreased, while Mclust ARIs decreased as the number of PCs increased. This reduction in classification accuracy aligns with the nature of the classification task in this dataset, which deals with more extensive spatial domains rather than finer cellular granularity, thereby indicating that broader spatial domains might be more appropriately captured with fewer PCs (Fig P in S1 Text).

## 2.13 Influence of $\beta$ on inverse distance weighting

To examine the effects of varying the parameter $\beta$ on inverse distance weighting, we computed a sparse distance matrix representing the spatial relationships among data points. Subsequently, we applied an inverse weighting transformation to this distance matrix, raising it to different powers of $\beta$. The relationship between distance (um) and the corresponding weights, calculated at $\beta$ values between 0 and 2, is visualized (Figs Q and R in S1 Text). A $\beta$ value of 0 results in a relatively uniform weighting across distances, effectively creating a rectangular kernel. In contrast, $\beta = 2$ increases the importance of small distances, and down weighing the importance of larger distances. The inverse distance weightings for $\beta = 0, 1, 2$ were applied to the first six PCs computed on the Mouse Ovary dataset and visualized spatially (Fig Q in S1 Text). Setting $\beta = 0$ results in large-scale smoothing across the structures present within the tissue. In contrast, $\beta = 2$ facilitates a localized smoothing effect, resulting in finer granularity of the observed structures. These results underscore the importance of $\beta$ in modulating the influence of neighboring points on the distance-weighted computations.

## 2.14 Differential distance metrics and weighting analysis

For the RASP results presented in Sects 2.3–2.8, spatial smoothing was performed using a sparse inverse distance matrix where each element is computed as the inverse Euclidean distance (raised to the power of $\beta$) between ST locations, with diagonal elements set to either the minimum distance (or half the minimum distance for Visium technology). We choose Euclidean distance for a number of reasons. First, the raw spatial coordinates generated by any imaging-based ST platform (Visium, Xenium, MERFISH, Stereo-Seq, etc.) are expressed in real-space units (microns), and Euclidean distance is the natural metric by which physical proximity on a 2D tissue section is measured. Second, Euclidean geometry is both isotropic and translation-invariant: cells or spots that are offset by the same vector are treated identically regardless of orientation, ensuring unbiased smoothing across the tissue. Third, from a computational standpoint Euclidean distances can be computed efficiently and are straightforward to threshold or sparsify—critical for scaling to high-resolution, subcellular datasets. The inverse distance weighting of the Euclidean distance captures the biologically plausible principle that the strength of spatial relationships decays continuously with physical separation, with the $\beta$ parameter allowing flexible modeling of interaction ranges, from highly local (large $\beta$) to more diffuse (small $\beta$).

To systematically evaluate the impact of both the distance metric and the weighting kernel, we evaluated Euclidean, Manhattan, and Chebychev distances as well as kernel-based weightings (Gaussian, quadratic), in addition to the standard inverse distance method, on both spot-based (DLPFC Visium) and an imaging-based (ovary MERFISH) ST datasets (Figs S and T in S1 Text). For each combination, we conducted clustering on the spatially smoothed principal components, and computed ARI relative to ground truth annotations, sweeping across a range of kNN and $\beta$ parameters (as well as kernel decay settings for the alternative weightings).

Across both datasets, our results demonstrate that no alternative metric or kernel provided a consistent or significant improvement over inverse Euclidean distance weighting. In all cases, ARI values were highly similar for different distance and kernel choices across the full parameter grid, indicating that the simpler and biologically motivated approach of inverse Euclidean weighting is both effective and robust. These findings justify our default choice of Euclidean distance and inverse distance weighting for RASP, balancing biological plausibility, computational efficiency, and empirical performance.

## 3 Discussion

Spatially informed dimensionality reduction of ST data is crucial for downstream tasks such as spatial domain identification, cell type annotation, and reduced rank gene signature reconstruction. In this study, we described RASP, a computationally efficient spatially informed dimensionality reduction method that seamlessly integrates into the standard analysis workflow provided by the Scanpy package. RASP utilizes randomized PCA followed by spatial smoothing to generate latent variables that can be leveraged for a wide range of downstream analyses including clustering to predict cell types or

 

spatial domains. RASP also facilitates the reduced rank reconstruction of spatially smoothed gene signatures, enhancing the interpretability of transcriptomic data in spatial contexts. An important feature of RASP is its support for the incorporation of non-transcriptomic covariates via an additional round of randomized PCA. Our findings demonstrate that RASP achieves similar, or higher, accuracy than existing methods for cell type and spatial domain detection tasks, with runtime performance that is orders-of-magnitude faster. RASP can flexibly accommodate various tissue types and ST technologies. Specifically, RASP accurately identifies biologically relevant tissue domains in structures that are homogeneous in size and shape, such as in the DLPFC dataset, while also providing robust labels akin to cell types in the highly heterogeneous mouse ovary sample. Notably, RASP is currently the only method among the techniques we evaluated that can efficiently handle the massive datasets produced by the 10x Xenium platform.

### 3.1 Parameter selection considerations

Several parameters require careful consideration when applying the RASP method to ST data. These include kNN threshold, $\beta$ value, and the number of PCs to compute. To simplify parameter selection, we provide a recommended range of values for each parameter. Please see Table 1 for details.

Note that we chose to use inverse distance weighting instead of the more commonly employed Gaussian kernel weighting given the more direct biological interpretation (e.g., inverse squared distances correspond to the concentration of a secreted molecule) and increased flexibility (i.e., users can control both the power parameter and diagonal values instead of just a bandwidth parameter). While RASP defaults to inverse distance weighting, the method can be used with alternative distance metrics and smoothing approaches, each with different characteristics and potential benefits/disadvantages.

### 3.2 Non-transcriptomic covariates

The RASP method allows for the incorporation of additional location-specific covariates via a two-stage randomized PCA (these are specified using the covariate matrix **Y** in Algorithm 1). Potential covariates include local cell density and cell volume (for technologies that provide this information), the library size of each location/cell, and protein abundance estimates (for technologies that support joint transcriptomic/proteomic measurements). However, it is important to note that using cell density may be somewhat redundant, as the kNN threshold inherently accounts for local density variations. However, depending on the cluster algorithm, this covariate can still impart useful information (see Fig 3B). Therefore, users should carefully select these parameters based on the specific objectives of their analysis. RASP's fast runtime allows users to quickly screen a large range of parameter values to identify the optimal configuration. RASP contains functionality to quickly calculate Moran's I and CHAOS scores on the resulting clusters and select the appropriate parameters for the final implementation. This is important because it allows users to leverage a priori knowledge to incorporate biologically relevant covariates that are not present in the transcriptomic data in a flexible manner, functionality other tools lack.

### 3.3 Limitations

While RASP has numerous advantages relative to existing spatially aware dimensionality reduction techniques, it is not without limitations. One of the primary concerns involves the specification of user-provided parameters. Although the default parameters are generally appropriate for a variety of situations, RASP's smoothing threshold parameter can profoundly influence clustering results. Some degree of trial and error is often necessary to optimize this parameter. However, unlike competing tools that may require upwards of 30 minutes to run, RASP executes quickly, facilitating a straightforward optimization process for users. To assist with parameter selection, we provide guidelines that can help streamline this aspect of the workflow (Table B in S1 Text). Additionally, it is important to note that for certain datasets, RASP may not be the best performing method. For instance, in the task of cortical layer assignment in the DLPFC, models such as

BASS outperform RASP. Additionally the RASP-based reduced rank gene reconstruction assumes that nearby cells have similar expression levels. However, this is not necessarily the case, and smoothing across tissue boundaries can occur when RASP's kNN parameter is set at a large value. We recommend that RASP-based reduced rank reconstruction be used primarily for visualization purposes.

## 3.4 Future directions

Multi-slice alignment and integration of sections from the same organism over different time points are two areas that have witnessed numerous computational innovations over the last five years. Methods such as MOFA, MEFESTO and moscot utilize matrix factorization-based models to simultaneously model both temporal and spatial data [60–62]. PALMO employs variance decomposition analysis and incorporates some spatial modeling capabilities [63]. In contrast, methods such as PASTE2 and DeST-OT utilize optimal transport principles to align sections from the same tissue sample, enabling 3D models [64,65]. Despite their promising capabilities, these models are computationally intensive, do not scale well to large datasets, and, importantly, lack streamlined approaches for joint dimensionality reduction and clustering. In the current study, we evaluated RASP's performance on single sections of ST data. Future directions include scaling RASP to support joint spatially informed dimensionality reduction of multiple ST slices, encompassing various tissues or multiple time points. While this enhancement can be simply realized using RASP by analyzing the concatenation of the cell-by-gene and coordinate matrices, how best to handle potential batch effects is a non-trivial problem. Given its computational efficiency, RASP stands out as the sole method capable of jointly reducing massive subcellular ST datasets. Another promising avenue of investigation involves integrating multi-modal information into the spatial reduction process. For instance, data related to protein abundance from methodologies like CITE-seq or chromatin accessibility assessed via spatial ATAC-seq could significantly enhance spatial domain detection [66,67]. RASP already includes functionality to support the incorporation of non-transciptomic covariates, and future research will focus on understanding the impact of these diverse data types in improving spatial domain detection and characterization.

## 4 Methods

### 4.1 RASP method

One of the primary motivations for spatially informed dimensionality reduction of ST data is the identification of spatial domains via unsupervised clustering. Clustering on standard PCs is often sufficient to identify cell types, but identification of larger tissue structures benefits from spatial information. The RASP method performs dimensionality reduction and spatial smoothing on normalized ST data and an optional set of location-specific covariates. The two-stage approach used by RASP was chosen to optimize computational efficiency and support the specification of covariates with an overall goal of enabling the flexible and efficient analysis of high-resolution ST datasets. Specifically, RASP uses a sparse matrix representation of the ST data, performs dimensionality reduction using randomized PCA, and generates spatially smoothed PCs using inverse distance weights sparsified according to a k-nearest neighbors (kNN) threshold (Fig 1A). To support the integration of location-specific covariates into the latent variables, a second stage randomized PCA is optionally performed on the matrix created by appending the smoothed covariates to the smoothed PCs generated in the first stage. The spatially smoothed PCs generated by the RASP method have utility in a wide range of downstream analyses with clustering and reduced rank reconstruction of gene-level expression important use cases. A detailed description of the method is provided by Algorithm 1. For readability, this version of the algorithm automatically applies the same spatial smoothing to both the first stage PCs and the covariates, however, flexible smoothing of the covariates is supported so that users have control over whether to apply smoothing to each covariate and, if applied, the type of threshold that is employed.

## Algorithm 1 RASP method.

**Inputs:**

- **X**: $n \times m$ location-by-gene matrix that holds normalized expression values for $n$ spatial locations and $m$ genes.
- **C**: $n \times 2$ matrix of spatial coordinates.
- $p$: Number of spatially smoothed PCs to compute.
- *threshold*: Threshold value. This is the number of nearest neighbors to retain.
- $\alpha$: The diagonal elements of the distance matrix are set to this value to avoid division by zero when computing inverse distances. Default value is the minimum off-diagonal distance for the row.
- $\beta$: Power used for inverse distance weighting. Default is 2.
- **Y**: Optional $n \times b$ matrix of $b$ location-specific covariates, e.g., local cell density. While spatial smoothing is automatically applied to these variables in the algorithm, flexible control over covariate smoothing is possible in practice.

**Outputs:**

- $\mathbf{P_s}$: $n \times p$ matrix that holds the spatially smoothed projection of **X** onto the top $p$ PCs.
- $\mathbf{Y_s}$: $n \times b$ matrix that holds the spatially smoothed version of **Y**.
- **W**: $m \times p$ matrix of loadings for the top $p$ PCs of **X**.
- $\mathbf{P_c}$: If **Y** is specified, the $n \times p$ matrix that holds the projection of $[\mathbf{P_s}, \mathbf{Y_s}]$ onto the top $p$ PCs where $[\mathbf{P_s}, \mathbf{Y_s}]$ is the $(p+b) \times n$ matrix formed by appending the columns of $\mathbf{Y_s}$ to $\mathbf{P_s}$.
- $\mathbf{W_c}$: if **Y** is specified, the $(p+b) \times p$ matrix of loadings for the top $p$ PCs computed on $[\mathbf{P_s}, \mathbf{Y_s}]$.

**Notation:**

- Let **W**[] represent a subsetting of the matrix **W** with $\mathbf{W}[i,j]$ the element in the $i^{th}$ row and $j^{th}$ column and with $\mathbf{W}[.,.]$ the iteration over all $i, j$ index combinations.

1: $\mathbf{W} \leftarrow randomizedPCA(\mathbf{X}, p)$ ▷ Perform randomized PCA on **X** for rank $p$. By default, **X** is centered but not scaled. See Algorithm A in S1 Text for details.
2: $\mathbf{P} \leftarrow \mathbf{XW}$ ▷ Project **X** onto the top $p$ PCs.
3: $\mathbf{NN} \leftarrow fitNearestNeighbors(\mathbf{C}, threshold)$ ▷ Fit a nearest neighbors data structure on the spatial coordinates for *threshold* neighbors.
4: $\mathbf{D} \leftarrow sparseMatrix(\mathbf{NN})$ ▷ Compute a sparse distance matrix using **NN**. See Algorithm D in S1 Text for details.
5: $diag(\mathbf{D}) \leftarrow \alpha$ ▷ Set diagonal elements of **D** to $\alpha$.
6: $\mathbf{D}_I[.,.] \leftarrow 1/\mathbf{D}[.,.]^{\beta}$ ▷ Create an inverse distance matrix using power $\beta$. Inverse distances for thresholded elements are set to 0.
7: $\mathbf{D}_I[.,.] \leftarrow \frac{\mathbf{D}_I[.,.]}{\max(\mathbf{D}_I[.,.])}$ ▷ Standardize each column of $\mathbf{D}_I$ to the maximum of that column.
8: $\mathbf{P_s} \leftarrow \mathbf{D}_I \mathbf{P}$ ▷ Spatially smooth the PCs using inverse distance weights.
9: **if Y then** ▷ If covariate matrix **Y** is specified.
10: $\mathbf{Y_s} \leftarrow \mathbf{D}_I \mathbf{Y}$ ▷ Spatially smooth covariates.
11: $\mathbf{W_c} \leftarrow randomizedPCA([\mathbf{P_s}, \mathbf{Y_s}], p)$ ▷ Perform randomized PCA on the merge of $\mathbf{P_s}$ and $\mathbf{Y_s}$.
12: $\mathbf{P_c} \leftarrow [\mathbf{P_s}, \mathbf{Y_s}] \mathbf{W_c}$ ▷ Project $[\mathbf{P_s}, \mathbf{Y_s}]$ onto the top $p$ PCs.
 **return** $\mathbf{P_s}, \mathbf{W}, \mathbf{Y_s}, \mathbf{P_c}, \mathbf{W_c}$
13: **else**
 **return** $\mathbf{P_s}, \mathbf{W}$
**end**

### 4.2 Clustering on spatially smoothed principal components

For evaluation of the RASP method, clustering was performed on the spatially smoothed PCs ($\mathbf{P_s}$ or $\mathbf{P_c}$) generated by RASP Algorithm 1. Specifically, clustering was performed by first computing a nearest neighbors distance matrix and a neighborhood graph of spatial locations as described in McInnes et al., 2018 [68]. See Sect 1.1 in S1 Text for programming language specific implementation. Note that the distances used for constructing this kNN graph are measured on the spatially smoothed PCs and capture transcriptomic differences between spatial locations, which is distinct from the

spatial distances between coordinates used in RASP Algorithm 1. Construction of the neighborhood graph requires specification of the number of nearest neighbors, with the recommended value being between 5 and 50. In general, the number of nearest neighbors should be set based on the size of the target spatial features. For example, users interested in cell type annotation are advised to set the threshold to a smaller value between 5 and 10, while those seeking to annotate spatial domains may consider increasing the threshold above 20. In most cases, a default of 10 is sufficient.

Distinct clusters representing spatial domains or cell types are then computed on the kNN graph using a user-defined clustering algorithm. The RASP method currently supports the use **Mclust**, **Louvain**, **Leiden**, and **Walktrap** methods. Note that for large datasets, users may want to avoid the use of the **Walktrap** clustering method as the computational complexity of the algorithm will increase runtime to an unreasonable level on a local machine.

## 4.3 RASP-based reduced rank reconstruction of gene-level expression

In addition to domain identification, an important application of spatially smoothed PCs is the reconstruction of gene-level expression values. Reduced rank reconstruction is motivated by the fact that the biological signal in transcriptomic data typically has a much lower rank than the rank of the observed measurements. Projecting ST data onto a lower dimensional space borrows the signal from correlated genes and can effectively mitigate both sparsity and noise [34]. For the spatially smoothed PCs generated by RASP Algorithm 1, the reconstruction process follows Algorithm 2. Important features of this method relative to standard PCA-based reduced rank reconstruction include the use of spatially smoothed PCs to incorporate information from adjacent locations, the optional reconstruction of the smoothed PCs to account for location-specific covariates, and the thresholding and optional scaling of the reconstructed values following the approach used by the ALRA method [34].

## 4.4 Application of RASP to existing ST datasets

### 4.4.1 Dataset descriptions.

We applied RASP on six publicly available ST datasets, as well as two simulated ST, that were generated using different techniques and resolutions on human and mouse tissues. Details of data preprocessing can be found in Sect 1.5 in S1 Text, details on data simulation can be found in Sect 4.4.4. In brief, raw count matrices where filtered to remove cells/locations with mitochondrial and ribosomal RNA > 10%, cells/locations with <100 reads and >5 times the mean absolute deviation (MAD). Resulting data were scaled, and log-normalized prior to running RASP (Sect 1.5 in S1 Text). The mouse ovary dataset (Vizgen MERFISH technology) consisted of 43,038 cells and 228 genes. The tissue section used in our analysis comes from a series of ST experiments profiling mouse ovaries at 0, 4 hr, and 12 hr after induced ovulation with human choronic gonadotropin (hCG). The tissue investigated in this method was from the 12 hr group and has subcellular resolution. Ground truth cell type annotations for eight cell types were taken from the original publication [15]. The mouse olfactory bulb dataset (STOmics Stereo-seq technology) consisted of 19,109 and 14,367 genes. This coronal section of the olfactory bulb is also profiled at sub-cellular resolution however the original publication did not provide ground truth regional annotations [49]. The human breast cancer tumor dataset (10x Genomics, Xenium technology) consisted of 565,916 cells and 541 genes, making it the largest dataset tested. This is a 5um thin resected infiltrating ductal carcinoma it situ with ground truth annotations provided in the **SubcellularSpatialData** R package [53]. Original annotations were determined from the accompanying histology image. The human DLPFC dataset (10x Genomics, Visium technology) consisted of 3,639 spots and 15,124 genes. The data here is a portion of the DLPFC that spans six neuronal layers plus white matter. The ground truth layer annotations come from the **spatialLIBD** R package [48].

The Sagittal adult mouse brain dataset (Vizgen MERFISH technology) consisted of 81,452 cells and 1,122 genes. The brain section is part of the Allen Institute's Mouse Brain Atlas originally described in [1,47]. The dataset includes region, cell class and cell type annotations. For ARI calculations we used the cell type level 'class', and division level 'parcellation division' labels. See Sect **??** for detailed instructions to access this data resource.

## Algorithm 2 Reduced rank reconstruction.

**Inputs:**

- **X**: $n \times m$ location-by-gene matrix that holds normalized expression values for $n$ spatial locations and $m$ genes. This is the same matrix provided as input to Algorithm 1.
- $\mathbf{P_s, W, P_c, W_c}$: Outputs from Algorithm 1. $\mathbf{P_c}$ and $\mathbf{W_c}$ are optional and should only be included if location-specific covariates were specified as inputs to Algorithm 1.
- $q$: Quantile probability used for thresholding reconstructed expression values.
- $scale$: True-to-scale reconstructed values to match variance of original expression.

**Outputs:**

- $\mathbf{X_r}$: Reduced rank reconstruction of $\mathbf{X}$.

**Notation:**

- Let $\mathbf{1}^{x \times y}$ be an $x \times y$ matrix of ones.
- Let $\mathbf{W}[]$ represent a subsetting of the matrix $\mathbf{W}$ with $\mathbf{W}[i,j]$ the element in the $i^{th}$ row and $j^{th}$ column, $\mathbf{W}[i,]$ the $i^{th}$ row, $\mathbf{W}[,j]$ the $j^{th}$ column, and $\mathbf{W}[\mathbf{r,c}]$ the sub-matrix containing rows with indices in $\mathbf{r}$ and columns with indices in $\mathbf{c}$.

1: **if** $\mathbf{P_c, W_c}$ **then** $\triangleright$ If $\mathbf{P_c}$ and $\mathbf{W_c}$ were specified
2: $[\mathbf{P_{s,r}, C_r}] \leftarrow \mathbf{P_c} \cdot \mathbf{W_c}^T$ $\triangleright$ Reconstruct smoothed PCs and covariates
3: $\bar{\mathbf{p}}_s \leftarrow (\mathbf{P_s}^T \mathbf{1}^{n \times 1})/n$ $\triangleright$ Compute column means of $\mathbf{P_s}$
4: $\mathbf{P_{s,r}} \leftarrow \mathbf{P_{s,r}} + \mathbf{1}^{n \times p} \bar{\mathbf{p}}_s^T$ $\triangleright$ Add back column means
5: $\mathbf{X_r} \leftarrow \mathbf{P_{s,r}} \mathbf{W}^T$ $\triangleright$ Reconstruct ST expression matrix using reconstructed PCs
6: **else**
7: $\mathbf{X_r} \leftarrow \mathbf{P_s} \mathbf{W}^T$ $\triangleright$ Reconstruct ST expression matrix using spatially smoothed PCs
 **end**
8: $\bar{\mathbf{x}} \leftarrow (\mathbf{X}^T \mathbf{1}^{n \times 1})/n$ $\triangleright$ Compute column means of $\mathbf{X}$
9: $\mathbf{X_r} \leftarrow \mathbf{X_r} + \mathbf{1}^{n \times m} \bar{\mathbf{x}}^T$ $\triangleright$ Add back column means
10: **for** $i \in \{1, ..., m\}$ **do** $\triangleright$ Threshold and optionally scale each reconstructed gene
11: $x_q \leftarrow quantile(\mathbf{X_r}[,i], q)$ $\triangleright$ Find quantile of reconstructed values for probability $q$
12: **for** $j \in \{1, ..., n\}$ **do** $\triangleright$ If absolute reconstructed value is greater than absolute quantile, keep, otherwise, replace with original value if that was positive or set to zero.

13: $\mathbf{X_r}[j,i] \leftarrow \begin{cases} \mathbf{X_r}[j,i] & \text{if } |\mathbf{X_r}[j,i]| \geq |x_q| \\ \mathbf{X}[j,i] & \text{if } \mathbf{X}[j,i] > 0 \text{ and } |\mathbf{X_r}[j,i]| < |x_q| \\ 0 & \text{if } \mathbf{X}[j,i] = 0 \text{ and } |\mathbf{X_r}[j,i]| < |x_q| \end{cases}$

 **end**
14: **if** scale **then**
15: $\sigma_i \leftarrow stddev(\mathbf{X}[,i] | \mathbf{X}[,i] > 0)$ $\triangleright$ Compute standard deviation of non-zero $\mathbf{X}[,i]$
16: $\sigma_{r,i} \leftarrow stddev(\mathbf{X_r}[,i] | \mathbf{X_r}[,i] > 0)$ $\triangleright$ Compute standard deviation of non-zero $\mathbf{X_r}[,i]$
17: $s \leftarrow \sigma_i / max(\sigma_{r,i}, 1e-10)$ $\triangleright$ Compute scaling factor (avoiding division by zero)
18: $\mathbf{X}[,i] \leftarrow s\mathbf{X_r}[,i]$ $\triangleright$ Apply scaling
 **end**
 **end**
 **return** $\mathbf{X_r}$

The Coronal adult mouse brain dataset (10x, Visium HD technology) consisted of 393,543 spots and 19,059 genes. This dataset does not contain expert or ground truth annotations.

The two simulated datasets (*Stripes*, and *Dots*) consisted of 10,000 cells and 150 genes each.

**4.4.2 RASP evaluation.** To evaluate the RASP method, we applied the algorithm to the processed gene expression matrices of the six datasets described above. The locations where then clustered according to the spatially smoothed PCs contained in either the $\mathbf{P_s}$ or $\mathbf{P_c}$ matrices output by RASP. Cluster quality was assessed using spatial continuity and compactness analysis quantified by the Moran's I value and CHAOS score (Sect 1.3 in S1 Text, Sect 1.4 in S1 Text) for

all data sets and, if ground truth annotations were available, the adjusted Rand index (ARI) (Sect 1.2 in S1 Text). The evaluation process required optimizing key parameters in the RASP pipeline, particularly the distance threshold used for constructing the sparse distance matrix and the $\beta$ value used to exponentiate the inverse distance matrix.

Given the large parameter space, a systematic parameter sweep was performed. To efficiently explore this space, we adopted a one-parameter-at-a-time optimization strategy. All other variables were held constant while one parameter incremented upon. Specifically, the number of kNN for spatial smoothing was swept from 1 to 50 for the MERFISH, Visium and Stereo-seq datasets, and from 1 to 120 for the Xenium and simulated datasets. Each kNN sweep was performed a total of nine times, for $\beta = 0, 0.25, 0.5, , 0.75, 1, 1.25, 1.5, 1.75, 2$. This approach allowed us to comprehensively explore the influence of each parameter individually. For each combination of parameters, the ARI (Sect 1.2 in S1 Text) was computed to quantify the similarity between the resulting clusters and the ground truth labels, where available, enabling the identification of the optimal parameter set for each dataset. For the olfactory bulb and Coronal mouse brain datasets ground truth annotations were unavailable and spatial continuity and compactness analysis was quantified by the CHAOS score, and the spatial spatial autocorrelation of each cluster was calculated using Moran's I score in place of an accuracy measurement.

**4.4.3 Comparison methods for spatial domain detection.** We compared the performance of RASP against seven other spatially informed dimensionality reduction or spatial domain detection algorithms: BASS (R, version 1.3.1), GraphST (Python, version 1.1.1), SEDR (Python, version 1.0.0), SpatialPCA (R, version 1.3.0), STAGATE pyG (Python, version 1.0.0), CellCharter (Python, version 0.3.5) NichePCA (Python, version 0.0.1), and MENDER (Python, 1.0.0) )[24, 26–30,32,33]. These methods where selected as they demonstrated the best performance across a variety of tissues and employ diverse model architecture. BASS models ST data in a hierarchical Bayesian framework and treats the cell type or spatial domain label for each spot or cell as a latent variable and infers them through an efficient inference algorithm. GraphST utilizes a GNN encoder to learn latent representations of the variables and further refines the representations using self-supervised contrastive learning. A decoder is employed to reconstruct gene expression, where clustering is employed. SEDR learns a low-dimensional representation of the data embedded with spatial information via a masked self-supervised deep auto-encoder and a variational graph convolutional auto-encoder. This low dimensional embedding is then clustered to identify spatial domains. SpatialPCA models gene expression as a function of latent factors through a factor analysis model. It builds a kernel matrix using the coordinates from the experiment to model the spatial correlation structure of the latent factors across the tissue. The inferred low-dimensional components are then clustered to identify spatial domains. STAGATE learns a low-dimensional latent representation of spatial information and gene expression via a graph attention auto-encoder trained on a spatial neighbor network (SNN). The output is then clustered to identify spatial domains. CellCharter is a domain detection framework that utilizes Variational Autoencoders (VAEs) to reduce dimensionality and remove batch effects, and then clusters the latent space using iterative Gaussian Mixture Modeling (GMM) to determine k clusters and cluster identities. NichePCA is similar to RASP, but aggregates log normalized gene expression counts through spatial averaging prior to running PCA. Downstream clustering is then applied in a similar manner to normal processing pipelines. Finally, MENDER uses cell context representation at scales to determine optimal neighborhood structures. The algorithm first constructs a spatial graph based on the cell coordinates, computed cell state frequencies across multiple ranges to form the cell context representation.

For each method, we followed the associated GitHub tutorials to run the stated methods for spatial domain and cell type detection, unless otherwise specified. Details are provided in Sect 1.7 in S1 Text. All methods where run on a Linux server with 25 CPUs for fair comparison of runtime. The latent dimensions found by each method were clustered using three clustering algorithms, Mclust, Louvain, and Leiden. The Walktrap clustering method was also used on the DLPFC dataset. Accuracy of the clusters generated by each method was determined by calculating ARI and CHAOS scores (Sect 1.2 in S1 Text, Sect 1.3 in S1 Text). Note that these methods required allocations of arrays > 1TB for the human breast cancer dataset and as such where unable to run successfully.

**4.4.4 Generation of simulated ST data.** We simulated ST data according to two different tissue models, *Stripes* and *Dots*, using the **SRTsim** package's GUI [59], following the reference free example provided at https://xzhoulab.github.io/SRTsim/03_Reference_Free_Example/. The *Stripes* model represents a laminar tissue with eight vertical domains (Fig J in S1 Text). The *Dots* dataset represents a more complex tissue architecture that includes circular clusters of cells of varying sizes, small striped regions, and a large proportion of background cells making up a background region (Fig K in S1 Text). Both models include eight annotated spatial domains. Count data was simulated according to these models for 10,000 cells, 50 low signal genes, 50 high signal genes and 50 noise genes using a zero inflated negative binomial distribution with $\mu = 2$, zero % = 0.5, and $\theta = 0.5$. Each annotated tissue region was assigned a unique fold change value (relative to $\mu$) to simulate differences in spatial domains seen in real tissue. We passed the *Stripes* and *Dots* models to the `reGenCountshiny` function with different *seed_number* values to generate 100 replicate count matrices for each. For model details, spatial coordinates and domain labels for the simulated datasets please refer to section "Data availability".

For details on how the simulated data was processed prior to running RASP see Sect 1.6 in S1 Text.

## Supporting information

**S1 Text. Includes supplemental methods and results.**
(PDF)

## Acknowledgments

Special thanks to Ruixu Huang for providing access to the mouse ovary dataset.

## Financial disclosure

The authors declare that they have no financial interests or relationships that could be construed as a potential conflict of interest related to this work.

## Author contributions

**Conceptualization:** Ian Gingerich, Hildreth Robert Frost.

**Data curation:** Ian Gingerich.

**Formal analysis:** Ian Gingerich.

**Funding acquisition:** Brittany A. Goods, Hildreth Robert Frost.

**Investigation:** Ian Gingerich.

**Methodology:** Ian Gingerich, Hildreth Robert Frost.

**Project administration:** Brittany A. Goods, Hildreth Robert Frost.

**Resources:** Hildreth Robert Frost.

**Software:** Ian Gingerich.

**Supervision:** Brittany A. Goods, Hildreth Robert Frost.

**Validation:** Ian Gingerich.

**Visualization:** Ian Gingerich.

**Writing – original draft:** Ian Gingerich.

**Writing – review & editing:** Ian Gingerich, Brittany A. Goods, Hildreth Robert Frost.

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
