## [Decision Letter · Decision Letter 0]

16 Sep 2025

PCOMPBIOL-D-25-01571

Randomized Spatial PCA (RASP): a computationally efficient method for dimensionality reduction of high-resolution spatial transcriptomics data

PLOS Computational Biology

Dear Dr. Gingerich,

Thank you for submitting your manuscript to PLOS Computational Biology. After careful consideration, we feel that it has merit but does not fully meet PLOS Computational Biology's publication criteria as it currently stands. Therefore, we invite you to submit a revised version of the manuscript that addresses the points raised during the review process.

Please submit your revised manuscript within 60 days Nov 16 2025 11:59PM. If you will need more time than this to complete your revisions, please reply to this message or contact the journal office at ploscompbiol@plos.org. Please include the following items when submitting your revised manuscript:

We look forward to receiving your revised manuscript.

Kind regards,

Jean Fan

Section Editor

PLOS Computational Biology

**Additional Editor Comments:**

The authors present a practically useful tool for scalable spatially aware dimensionality reduction RASP. In order to warrant publication, the current paper should be revised to include more comparison with existing tools to better clarify the unique attributes of RASP, application to additional spatial transcriptomic datasets with larger gene panels to demonstrate scalability and applicability to data from newer spatial technologies, as well as additional clarifications on hyperparameters to guide users.

**Journal Requirements:**

At this stage, the following Authors/Authors require contributions: Ian Gingerich, Brittany A Goods, and H. Robert Frost. Please ensure that the full contributions of each author are acknowledged in the "Add/Edit/Remove Authors" section of our submission form.

5) We have noticed that you have uploaded Supporting Information files, but you have not included a list of legends. Please add a full list of legends for your Supporting Information files after the references list.

7) Please provide a completed 'Competing Interests' statement, including any COIs declared by your co-authors. If you have no competing interests to declare, please state "The authors have declared that no competing interests exist".

**Reviewers' comments:**

Reviewer's Responses to Questions

**Comments to the Authors:**

**Please note that one review is uploaded as an attachment.**

Reviewer #1: I uploaded the review as an attachment.

Reviewer #2: This manuscript presents RASP, a scalable spatially aware dimension reduction method that integrates randomized PCA with post-hoc KNN inverse-distance smoothing. The method is computationally efficient, and can easily fit into mainstream ST workflows. This method's novelty is incremental but practically useful. The authors evaluated the results across multiple platforms (Visium, MERFISH, Stereo-seq, Xenium), and the tool appears pretty practical. Below are some comments I have:

1. RASP is a dimensiona reduction method, not a clustering or domain detection method itself. The cell types and spatial regions are obtained from its output spatially smoothed PCs through clustering, and the resulting cluster resolution depends on KNN and β parameters. This is described in the methods and results of the manuscript, but should be made explicit in the Abstract and Introduction to avoid confusion. Currently it reads like this is a method for spatial region detection in abstract.

2. On line 99, BASS is listed as a "spatial domain detection" method. However, BASS performs joint cell type and domain inference. Additionally, in Fig. 4A (mouse brain), where comparisons are made on cell type classification, it would be more appropriate to use the cell-type labels produced by BASS.

3. Gene expression reconstructions are only validated qualitatively. The current manuscript lacks quantitative evaluation on gene expression reconstruction. The authors could consider either predict masked gene expression counts at locations through MSE or correlation, or held-out a subset of genes and quantify reconstruction accuracy.

4. The manuscript shows parameter sensitivity analysis and claims that users can tune KNN and β using metrics like Moran’s I and CHAOS. To improve usability, please provide practical defaults and a recommendation table for typical KNN and β values for cell-type annotation vs. spatial domain detection,

or preferred clustering algorithms (e.g., Louvain, Leiden), and suggestions for choosing cluster number or resolution in unsupervised settings.

**Have the authors made all data and (if applicable) computational code underlying the findings in their manuscript fully available?**

Reviewer #1: Yes

Reviewer #2: Yes

PLOS authors have the option to publish the peer review history of their article (what does this mean?). If published, this will include your full peer review and any attached files.

Reviewer #1: No

Reviewer #2: No

**Figure resubmission:**
---

## [Decision Letter · Decision Letter 1]

17 Nov 2025

Dear Mr Gingerich,

We are pleased to inform you that your manuscript 'Randomized Spatial PCA (RASP): a computationally efficient method for dimensionality reduction of high-resolution spatial transcriptomics data' has been provisionally accepted for publication in PLOS Computational Biology.

Best regards,

Jean Fan

Section Editor

PLOS Computational Biology

Jean Fan

Section Editor

PLOS Computational Biology

Reviewer's Responses to Questions

**Comments to the Authors:**

Reviewer #1: I am satisfied with the manuscript in its revised form.

Reviewer #2: The reviewers have addressed all my comments.

**Have the authors made all data and (if applicable) computational code underlying the findings in their manuscript fully available?**

Reviewer #1: Yes

Reviewer #2: Yes

PLOS authors have the option to publish the peer review history of their article (what does this mean?). If published, this will include your full peer review and any attached files.

Reviewer #1: No

Reviewer #2: No

---

## [Editor Report · Acceptance letter]

PCOMPBIOL-D-25-01571R1

Randomized Spatial PCA (RASP): a computationally efficient method for dimensionality reduction of high-resolution spatial transcriptomics data

Dear Dr Gingerich,

I am pleased to inform you that your manuscript has been formally accepted for publication in PLOS Computational Biology. Your manuscript is now with our production department and you will be notified of the publication date in due course.

With kind regards,

Judit Kozma
